# Upregulation of angiotensin-(1–7) formation in human podocytes – enzyme activity assay upon fluid flow shear stress

Debora Kaiser-Graf[1]*, Oliver Domenig[2], Marko Poglitsch[2], Reinhold Kreutz[1], Juliane Bolbrinker[1]

**1** Charité – Universitätsmedizin Berlin, corporate member of Freie Universität Berlin and Humboldt-Universität zu Berlin, Institute of Clinical Pharmacology and Toxicology, Berlin, Germany, **2** Attoquant Diagnostics GmbH, Vienna, Austria

* debora.kaiser@charite.de

## Abstract

The renin-angiotensin system is known for its role in renal physiology by regulating renal hemodynamics and natriuresis. Overactivation of this system exerts pathological effects in the kidney, primarily mediated by its main effector peptide angiotensin II. Deleterious angiotensin II-mediated effects are counter-regulated at least partly by the biologically active renin-angiotensin system component angiotensin-(1–7). The processing of angiotensin II to angiotensin-(1–7) by prolylcarboxypeptidase, angiotensin converting enzyme 2, and prolyl endopeptidase is cell- and species-specific, with limited knowledge regarding its conversion in human podocytes under conditions of glomerular hyperfiltration. Since hyperfiltration plays a critical mechanistic role in albuminuria progression and chronic kidney disease, understanding the mechanisms underlying podocyte damage due to glomerular hyperfiltration is essential. Therefore, we investigated the conversion of angiotensin II to angiotensin-(1–7) in cultured human podocytes exposed to fluid flow shear stress and subsequently incubated with spiked angiotensin II. Mass spectrometry of cell lysates and supernatants was performed to evaluate the formation of angiotensin-(1–7). Contribution of the respective enzymes to angiotensin-(1–7) formation was assessed using selective inhibitors of prolylcarboxypeptidase, angiotensin converting enzyme 2, and prolyl endopeptidase. We detected increased angiotensin-(1–7) formation upon fluid flow shear stress in podocyte lysates, which was mainly dependent on prolylcarboxypeptidase activity. Our study contributes to a deeper understanding of the intraglomerular processing of angiotensin II towards the alternative renin-angiotensin system and its modulation upon fluid flow shear stress.

**Data availability statement:** All relevant data are within the manuscript and its Supporting Information files.

**Funding:** The author(s) received no specific funding for this work.

**Competing interests:** Oliver Domenig and Marko Poglitsch had a paid employment at Attoquant Diagnostics GmbH within the last 5 years. This commercial affiliation does not alter our adherence to PLOS ONE policies on sharing data and materials.

## Introduction

The renin–angiotensin system (RAS) is a key system for regulation of renal hemodynamics and homeostasis [1,2]. Overactivation of RAS leads to various intrarenal hemodynamic and cellular alterations. These changes include efferent arteriolar vasoconstriction, elevated intraglomerular pressure, and glomerular hyperfiltration (GH), potentially leading to a decline in kidney function [3–7]. GH plays a crucial mechanistic role in the progression of albuminuria and the worsening of chronic kidney disease (CKD) in the context of hypertension and diabetes [8–11]. Several studies emphasize the critical role of angiotensin II (Ang II) in both physiological and pathological conditions in the kidney including vasoconstriction, growth and fibrosis, inflammatory response, and oxidative stress, thereby modulating renal hemodynamics [12–16]. Angiotensin II receptor type 1 (AGTR1) mediates the majority of the pathophysiological effects of Ang II in the kidney, while Angiotensin II receptor type 2 (AGTR2) antagonizes several of the actions mediated by AT1R [17]. Upregulation of renal Ang II-mediated effects are associated with increased blood pressure and play a central role in the development and maintenance of hypertension independent from changes in circulating RAS [18,19].

Adaptation of glomerular hemodynamics results in an elevated glomerular filtration rate (GFR) and concomitantly higher ultrafiltrate flow within Bowman's space, leading to increased fluid flow shear stress (FFSS) at the podocyte surface and to podocyte damage [20–25]. Applying FFSS on podocytes using a shear stress device represents an in vitro approach to mimic the increased ultrafiltrate flow in Bowman's space as observed upon GH, including cellular responses such as actin cytoskeleton rearrangement and morphological alterations that contribute to podocyte injury [23–29]. Podocytes are fully differentiated epithelial cells that form the third layer of the glomerular filtration barrier and are damaged in most proteinuric kidney diseases [10,30,31]. During disease progression, albuminuria often correlates with markers of podocyte injury including foot process effacement, podocyte hypertrophy, and apoptosis [32–34]. The crucial role of Ang II in podocyte injury and the progression of diabetic and non-diabetic kidney disease has been extensively investigated [31,35–40]. Podocytes are directly affected by Ang II–mediated injury through altered expression and distribution of podocyte proteins [31,41,42]. Furthermore, Ang II indirectly contributes to podocyte damage by enhancing calcium influx and reactive oxygen species production, which leads to actin cytoskeleton disruption, filtration barrier dysfunction as well as apoptosis [38,41,43].

Ang II is further metabolized to angiotensin-(1–7) [Ang-1–7] by prolyl endopeptidase (PREP; EC 3.4.21.26), prolylcarboxypeptidase (PRCP; EC 3.4.16.2), and angiotensin converting enzyme 2 (ACE2; 3.4.17.23)) [12,17,44–46]. Ang-(1–7) is also directly formed through hydrolysis from angiotensin I (Ang I) by PREP and membrane metalloendopeptidase (neutral-endopeptidase, nephrilysin) [12,44,47–49]. The Ang-(1–7) peptide is known to be a biologically active component of the RAS, mediating signaling through the MAS1 proto-oncogene receptor (MAS1), AGTR2, and MAS related GPR family member D receptor (MrgD) [13,50].

The ACE2-Ang-(1–7)-MAS1 axis is considered a counter-regulatory part of the ACE-Ang II-AGTR1 axis within the RAS as it modulates Ang II-mediated pathomechanisms [50]. In the kidney, Ang-(1–7) promotes vasodilation, inhibits growth, evokes anti-inflammatory responses and exerts antihypertensive and antifibrotic effects through the production of cAMP and nitric oxide (NO), activation of protein tyrosine phosphatases (PTPs), and inhibition of mitogen-activated protein kinases (MAPKs) and NADPH oxidases [6,12,13,46,51].

Reduced ACE2 activity, and consequently decreased Ang-(1–7) production, is associated with vasoconstriction, vascular remodeling, and oxidative injury in hypertension, diabetes, and kidney disease [47,52–55]. Therefore, activation of components of the ACE2-Ang-(1–7)-MAS1 axis are considered potential therapeutic targets [16,41,53].

Understanding the mechanisms and pathways underlying podocyte damage due to GH is essential for identifying potential new therapeutic targets. Thus, our objective was to clarify the regulation of the alternative and potentially renoprotective Ang-(1–7) formation in the context of elevated FFSS in human podocytes.

## Methods

### Cell culture

Conditionally immortalized human podocytes (hPC) [56,57] were kindly provided in June 2019 by Moin A. Saleem (University of Bristol, UK) and were cultured as previously described [25]. Briefly, for proliferation cells were cultivated at 33 °C and 5% $CO_2$ until 70–80% confluency in Roswell Park Memorial Institute (RPMI)-1640 medium (cat. no. BS.F1295, Bio&SELL, Feucht/ Nürnberg, Germany) supplemented with 1% Insulin-Transferrin-Selenium 100X (cat. no. 41400–045, Gibco, Grand Island, NY, US), 10% fetal bovine serum (FBS) (cat. no. F7524, Sigma, St. Louis, MO, US) and 1% Zell-Shield˚ (cat. no. 13–0150, Minerva Biolabs, Berlin, Germany). Cells were transferred to 38 °C and cultured until full confluency and proliferation arrest. For differentiation, cells were further cultivated at 38°C for another 14 days. Differentiated hPC phenotype was confirmed previously by detecting podocyte specific markers synaptopodin and nephrin on protein level [25].

### RNA isolation and reverse transcription

Total RNA of hPC was isolated and DNase-treated using the RNeasy® Micro Kit (cat. no. 74004, Qiagen, Hilden, Germany) according to the manufacturer's protocol. RNA quality was checked by 260/280 nm absorption ratio. cDNA synthesis was carried out on 2 µg of total RNA using the First Strand cDNA Synthesis Kit (cat. no. K1612, Thermo Fisher Scientific, Vilnius, Lithuania) following the manufacturer's protocol. Expression analysis of RAS components in hPC was determined by polymerase chain reaction (PCR) (S1 File).

### Immunofluorescence

Ang-(1–7) forming enzymes ACE2, PRCP and PREP were determined on protein level in hPC by immunofluorescence. Cells were seeded at 25 x 10³ cells per well in 8-chamber NUNC Lab-Tek II Chamber slides (cat. no. 154534, NUNC, Rochester, NY, US) and cultured for 48 h in RPMI-1640 medium with supplements for adherence. Medium was discarded and cells washed twice with phosphate buffered saline (PBS, cat. no. 14190094, Thermo Fisher Scientific, Waltham, MA, US). Cells were fixed with methanol (cat. no. 8045, JK.T. Baker, Deventer, Netherlands) for 5 min at −20 °C. Cells were then allowed to dry completely before being rehydrated and washed three times with PBS. Blocking solution consisting of PBS with 2% FBS was applied for 30 min at room temperature. Cells were incubated with primary antibody polyclonal rabbit anti-ACE2 (AB_2792286, cat.no. PA5–85139, Invitrogen, Rockford, IL, US) diluted 1:100, polyclonal rabbit anti-PRCP (AB_1855676, cat. no. HPA 017065, BIOZOL, Eching, Germany) diluted 1:1000, or polyclonal rabbit anti-PREP (AB_2900433, PA5–115798, Invitrogen, Rockford, IL, US) diluted 1:200 for 1 h at room temperature. After three washing steps, secondary antibody polyclonal goat anti-rabbit IgG Alexa Fluor™ Plus 488 (AB_2633280, cat. no. A32731,

Invitrogen, Rockford, IL, US) diluted 1:2000 was added and incubated for 1 h at room temperature in darkness. Nuclei were stained with 4',6-diamidino-2-phenylindole (DAPI, cat. no. D1306, Life Technologies, Carlsbad, CA, US) diluted 1:5000 for 10 min in darkness. Cells were washed in PBS and covered with Vectashield® mounting medium (cat. no. H-1000, Vector Laboratories, Burlingame, Ca, US). Immunofluorescence was detected at 405 nm (DAPI), and 488 nm (goat anti-rabbit for ACE2, PRCP and PREP) on a Zeiss axiovert 200 microscope (Carl Zeiss Microscopy GmbH, Jena, Germany) using ZEN blue v.3.4.91.0 software (Carl Zeiss Microscopy GmbH, Jena, Germany). Additionally, ACE2 and PRCP were determined on protein level in hPC by western blot (S1 File).

## Inhibition of Ang-(1–7) formation by different inhibitors

Ang-(1–7) formation from substrate Ang II by ACE2, PRCP, and PREP was investigated in the presence of five different inhibitors. Inhibition of enzyme activity was tested for the ACE2 inhibitor MLN-4760 (ACE2i) (cat. no. 305335-31-3, Sigma-Aldrich, St. Louis, MO, US), the PRCP inhibitor PRCPi (cat. no. 5.04044.0001, calbiochem, San Diego, CA, US), PREP inhibitor S17092 (PREPi$_1$) (cat. no. SML0181, Sigma-Aldrich, Saint Louis, MO, US), PREP inhibitor KYP-2047 (PREPi$_2$) (cat. no. 6272, Tocris Bioscience, Bristol, UK), and the dual PREP and PRCP inhibitor Z-Pro-Prolinal (dual PREPi + PRCPi) (ZPP, cat. no. 88795-32-8, Sigma Aldrich, St. Louis, MO, US). 10 ng/mL recombinant human PRCP (cat. no. 7164-SE, R&D Systems, Minneapolis, MN, US), 10 ng/mL recombinant human PREP (cat. no. 4308-SE, R&D Systems, Minneapolis, MN, US), and 10 ng/mL recombinant human ACE2 (cat. no. 933-ZN, R&D Systems, Minneapolis, MN, US) were incubated in presence of increasing ACE2i-, PRCPi-, PREPi$_1$-, PREPi$_2$-, and dual PREPi + PRCPi-concentrations [0.01 µmol/L – 100 µmol/L] for 1 h at 37 °C. Inhibitor solutions (pH 7.4) consisted of 10 µmol/L amastatin (cat. no. 4025800, Bachem, Bubendorf, Switzerland) and 10 µmol/L lisinopril (cat. no. L6292, Sigma-Aldrich, St. Louis, MO, US), as well as 5 mmol/L ZnCl$_2$ (cat.no. 703516, Sigma-Aldrich, St. Louis, MO, US) and 1 µg/mL Ang II as substrate (A9525, Sigma-Aldrich, St. Louis, MO, US) (pH 7.4). Subsequently, adjustments in the experimental setting were made for the inhibitory effect of ACE2i on ACE2 by lowering the amount of recombinant ACE2–1 ng/mL and for the inhibitory effect of PRCPi on PRCP by lowering the PRCPi concentration to 0.01 nmol/L – 100 nmol/L. Inhibitor concentrations were log-transformed and data are plotted as Ang-(1–7) formation [ng/mL] over inhibitor concentrations. Log (IC$_{50}$) values, IC$_{50}$ values, and R-squared values (R$^2$) were calculated using a non-linear regression (curve fit) of log(inhibitor) vs. response using GraphPad Prism 6.07 (GraphPad Software, San Diego, CA, US).

## FFSS

For FFSS, cells were detached with Trypsin 0.25%/EDTA 0.02% solution (cat. no. P10-023100, PAN-Biotech, Aidenbach, Germany). $3.5 \times 10^5$ cells were seeded on collagen IV coated Culture Slips® (cat. no. CS-C/IV, Dunn Labortechnik GmbH, Asbach, Germany) and adhered for 48 hours. FFSS experiments were performed as previously described with minor changes [25]. Briefly, Culture Slips® with hPC were inserted in the Streamer® Shear Stress Device (cat. no. STR-400, Dunn Labortechnik GmbH, Asbach, Germany) and the system was placed in the incubator at 38 °C with 5% CO$_2$. The Streamer® Shear Stress Device ensured a constant medium flow over the hPC surface the entire time. For the experiments, FFSS at 1 dyne/cm$^2$ for 2 h was applied based on previous research [21,24,27]. Control cells were cultured in the incubator simultaneously without FFSS exposure. All experiments were performed in supplement-free RPMI-1640 medium with cell passages between 7 and 15.

For determination of Ang-(1–7) in cell lysates, cells were released from the device at the end of the experiments, washed twice with PBS and stored immediately at −80°C until further use. For the Ang-(1–7) formation analyses in hPC supernatants, cells were removed from the device, washed with 38 °C supplement-free RPMI-1640 medium and processed immediately. Experiments were conducted twice on different cell passages and different dates, each time with n = 6 replicates for cell lysates analyses and n = 4 replicates for analyses of supernatants, respectively.

## Quantitative real-time PCR

Quantitative Real-Time PCR (qPCR) analyses were performed in CFX96™ Real-Time system (Bio-Rad, München, Germany, software version 3.1.1517.0823) using the comparative quantitative cycle method as reported [58,59]. Expression analysis of each sample was done in three technical replicates and only samples with an intra-triplicate standard deviation (SD) < 0.2 were used for further calculation. We assessed mRNA expression of *ACE2* and *PRCP*. Primer sequences are listed in S1 Table. Normalization of expression data was done by the reference gene glucuronidase beta (*GUSB*). All experiments for qPCR analyses were performed in quadruples, each time with n = 4–6 replicates.

## Determination of Ang-(1–7) formation in hPC upon FFSS

**Cell lysates sample preparation.** Ang-(1−7) formation was determined in presence and absence of ACE2i, with and without concomitant PRCPi in hPC lysates after applying FFSS. Cells were scratched from slides and transferred to a vial containing PBS. Cell suspensions were sonicated for dissolution using Vibra-Cell™ (Sonics & Materials Inc., Newtown, CT; US) and centrifugated at 16260 rcf for 10 min at 4°C. Supernatants were collected and normalized to total protein concentration determined using BIO-RAD Protein Assay Dye with bovine serum albumin as protein standard (cat. no. 5000006, Hercules, CA, US). Samples normalized to 200 µg total protein/mL were diluted in 100 µL inhibitor solution containing Ang II as substrate (Methods 4, see above) +/- 10 µmol/L ACE2i and +/- 10 µmol/L PRCPi and incubated for 1 h at 37 °. Thereafter, samples were acidified for stabilization using formic acid (10%, cat. no. 64-18-6, Sigma-Aldrich, St. Louis, MO, US).

**Cell supernatants sample preparation.** Ang-(1–7) formation rate on hPC surface was determined in presence and absence of ACE2i or PRCPi after applying FFSS. For this, hPC covered slides were inserted into 8-chamber casettes (Visium Casette Assembly, cat. no. 1000388, 10x Genomics, Pleasanton, CA; US) directly after FFSS experiments enabling segmentation of the slides. Inhibitor solutions containing substrate Ang II (Methods 4, see above) +/- 10 µmol/L ACE2i or 10 µmol/L PRCPi were added to supplement-free RPMI-1640 medium. 100 µL inhibitor solutions were applied each into one chamber of the cassette onto hPC and incubated for 1 h at 37 °C. Supernatants were collected and, following stabilization by acidification, Ang-(1–7) formation was quantified by LC-MS/MS (Methods 7.3).

**LC-MS/MS.** Stable isotope-labeled internal standards for Ang II and Ang-(1–7) were added to acidified samples at 200 pg/mL. Following C18-based solid-phase extraction, samples were subjected to LC-MS analysis using a reversed-phase analytical column (Acquity UPLC C18; Waters, Milford, Massachusetts) operating in line with a XEVO TQ-S triple quadrupole mass spectrometer (Waters) in MRM mode. Internal standards were used to correct for matrix effects and peptide recovery of the sample preparation procedure for both angiotensin peptides in each individual sample. Angiotensin peptide concentrations were calculated considering the corresponding response factors determined in appropriate calibration curves, on condition that integrated signals exceeded a signal-to-noise ratio of ten. Ang II concentration was measured in a non-incubated control sample as reference and after incubation in all samples to confirm that the substrate is not exhausted following incubation. Assay controls using recombinant enzymes were included to ensure that the inhibitor concentration was functional and to exclude non-specific formation of Ang-(1–7). Ang-(1–7) formation rate for cell lysates is expressed as (fmol/ µg protein)/ h and for supernatants as (nmol/ L)/ h. In cell lysates, the sole effect of PRCP inhibition on the Ang-(1–7) formation rate was calculated by subtracting remaining Ang-(1–7) formation rate upon ACE2 inhibition in each sample from the respective total value. This value was then summed to the ACE2i + PRCPi value of the respective sample which finally represents the individual PRCPi value.

## Statistics

Statistical analysis was performed using GraphPad Prism 6.07. Data are presented as mean ± SD, and $p < 0.05$ was considered as statistically significant. Normal distribution was tested with the Shapiro-Wilk test. Normally distributed data were compared by either unpaired, two-tailed Student`s t-test or one-way ANOVA with Tukey`s multiple comparison test. Data

not normally distributed were analyzed by Mann-Whitney test as indicated. For identification of outliers, Grubbs' outliers test ($\alpha = 0.05$) was performed.

## Results

### Characterization of angiotensin system components in hPC

Characterization of angiotensin system components in differentiated hPC revealed no angiotensinogen (*AGT*) and no renin (*REN*) gene expression (S1 Fig, panel A and B), whereas *ACE*, *ACE2*, *PRCP*, and *PREP* as well as *AGTR1* gene expression was observed (S1 Fig, panel C-G). *AGTR2* was only weakly and inconsistently expressed (S1 Fig, panel H). Immunofluorescence revealed expression of ACE2, PRCP, and PREP on protein level in hPC (Fig 1). Additional western blot analyses confirmed ACE2 and PRCP expression on protein level in hPC (S4 Fig, panel A-C).

### Enzymes involved in Ang-(1–7) formation from substrate Ang II

To determine the contribution of the three enzymes ACE2, PRCP, and PREP reported to generate Ang-(1–7) from Ang II on Ang-(1–7) formation in hPC, we first investigated the specificity of five different enzyme inhibitors. Enzyme activities of ACE2, PRCP, and PREP were determined by Ang II to Ang-(1–7) conversion with increasing inhibitor concentrations (Fig 2 and S2 Fig).

ACE2i was the only inhibitor blocking ACE2 activity with a remaining Ang-(1–7) formation detected at the highest ACE2i-concentration tested (Fig 2A). Using 1 ng/mL ACE2, complete inhibition of Ang-(1–7) formation was achieved ($IC_{50} = 508.2$ nM, $R^2 = 0.9994$; Fig 2B).

PRCPi had the strongest inhibitory effect on Ang II to Ang-(1–7) conversion by PRCP as 10 nM PRCPi already completely blocked the PRCP activity (Fig 2C). Thus, lower PRCPi concentrations (0.01 nM – 100 nM) were tested. Ang-(1–7) formation by PRCP was effectively inhibited by PRCPi with $IC_{50} = 78.36$ pM ($R^2 = 0.9997$; Fig 2D). The dual PREPi+PRCPi as well as $PREPi_2$ also had inhibitory effects on PRCP-derived Ang-(1–7) formation with $IC_{50}$ of 228.4 nM and 8336 nM respectively ($R^2 = 0.9994$, $R^2 = 0.9779$; S3 Fig, panel B).

We observed no PREP-mediated Ang-(1–7) conversion of 1 ng/mL (S2 Fig, panel A). Thus, a higher (1 µg/mL) PREP enzyme concentration was tested to ensure that PREP is not involved in the formation of Ang-(1–7) from Ang II. Even with this high enzyme concentration, only a very weak PREP-activity and no inhibitory effects were detected (S2 Fig, panel B; S3 Fig, panel D).

Overall, the total amount of Ang-(1–7) formation was 6-times higher with ACE2 (mean 519 ng/mL) compared to PRCP (mean 81 ng/mL) indicating ACE2 as the most effective enzyme in Ang-(1–7) formation. Inhibitor assays revealed that Ang-(1–7) formation from the substrate Ang II was mediated by ACE2 and PRCP, but not by PREP.

### Ang II to Ang-(1–7) conversion in hPC upon FFSS

Upon FFSS, there was a significantly elevated Ang II to Ang-(1–7) conversion rate in cell lysates of hPC with $720 \pm 111$ (fmol/µg protein)/h compared to no flow controls $554 \pm 87$ (fmol/µg protein)/h, $p = 0.0007$ (Fig 3A). In contrast to elevated Ang-(1–7) formation in cell lysates, no effect was observed on Ang-(1–7) formation rate at the cell surface of hPC (Fig 3B). These results suggest a higher enzyme activity and Ang II to Ang-(1–7) conversion capacity upon FFSS in hPC.

Gene expression analysis revealed a slightly upregulated *ACE2* expression upon FFSS exposure compared to controls ($p < 0.05$; Fig 4A). In contrast, *PRCP* expression levels were slightly reduced upon FFSS compared to controls ($p < 0.05$; Fig 4B). Overall the enzymes involved in Ang II to Ang-(1–7) conversion were only moderately regulated upon FFSS.

### Contribution of ACE2 and PRCP for Ang-(1–7) formation in hPC upon FFSS

**Cell lysate.** To determine the proportion of ACE2 and PRCP involved in Ang-(1–7) formation rate from Ang II, hPC lysates were incubated with and without ACE2i and PRCPi. After FFSS exposure as well as under no flow control

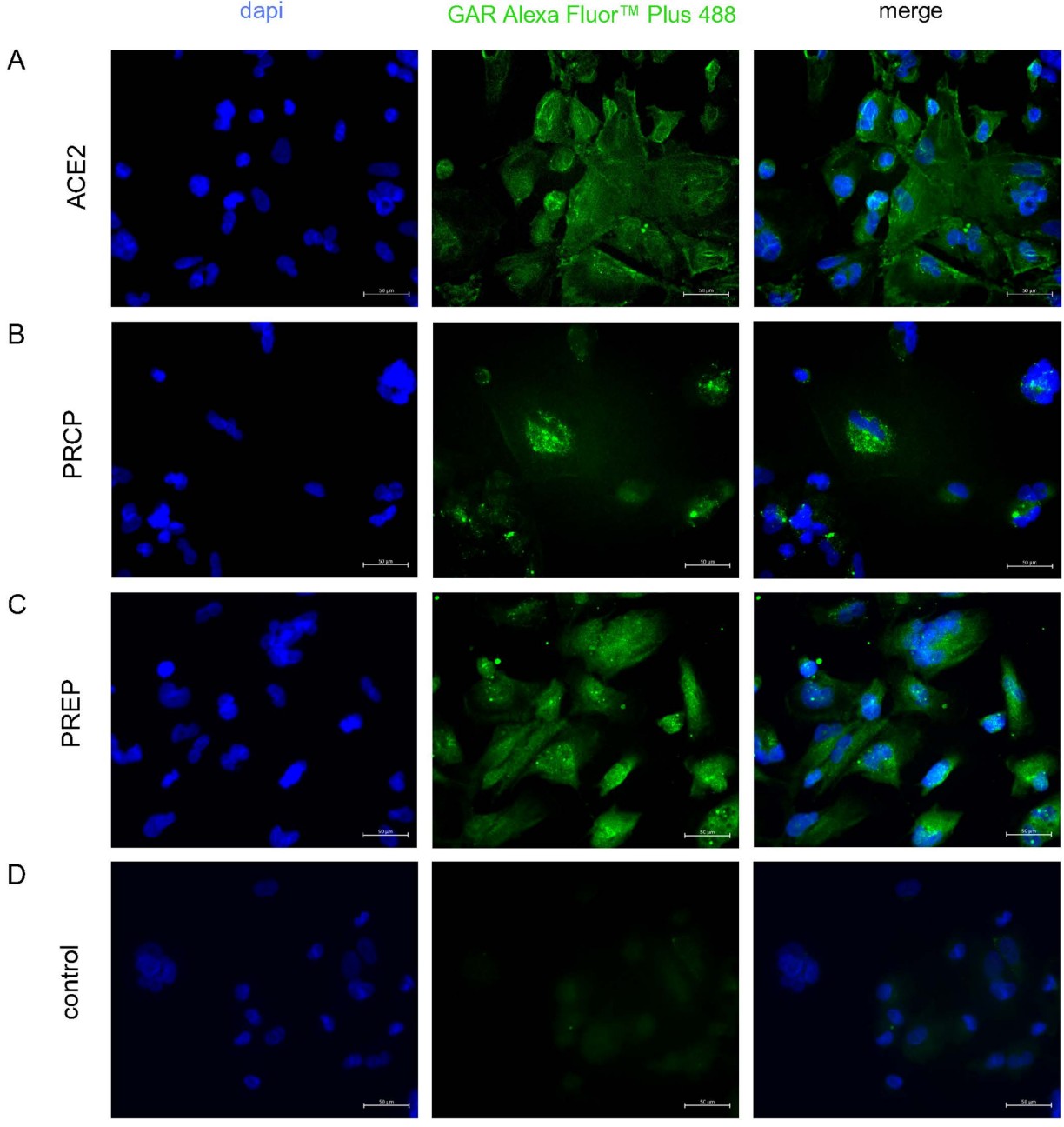

**Fig 1. Characterization of ACE2, PRCP, and PREP on protein level in hPC conducted by immunofluorescence.** Representative immunofluorescence images of **(A)** ACE2, **(B)** PRCP, and **(C)** PREP, demonstrating protein expression on hPC. **(D)** Control without primary antibody revealed no specific binding of secondary antibody goat anti-rabbit Alexa Fluor™ Plus 488. Nuclei were stained using DAPI (blue). Scale bar 50 μm.

conditions, inhibition of ACE2 did not affect the Ang-(1–7) formation rate in hPC lysates compared to the total Ang-(1–7) formation rate. Inhibition of PRCP resulted in a significantly reduced Ang-(1–7) formation rate by at least half regardless of FFSS exposure or no flow control (p < 0.0001; Fig 5). Despite ACE2 and PRCP inhibition, a residual Ang-(1–7) formation rate remained, which was even higher under FFSS conditions compared to controls. However, Ang-(1–7) formation in hPC

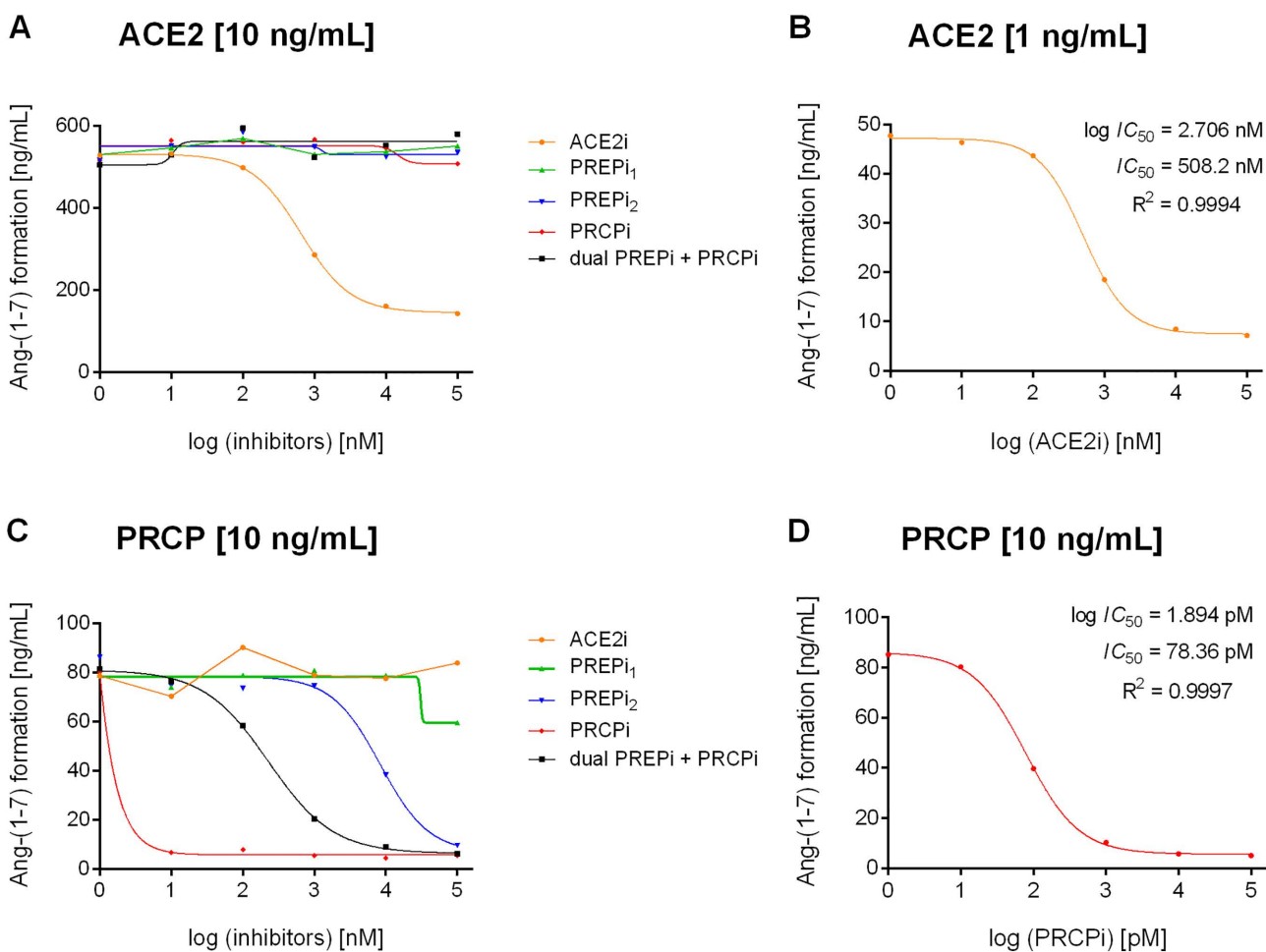

**Fig 2. Inhibitor effects on ACE2-, PRCP-, and PREP-mediated Ang II to Ang-(1-7) conversion. (A)** Inhibition of Ang-(1-7) formation [ng/mL] by ACE2 (10 ng/mL) was tested with increasing inhibitor concentrations of five different inhibitors. **(B)** For a complete inhibition of Ang-(1-7) formation by ACE2, 1 ng/mL ACE2 was incubated with increasing ACE2i concentrations [0.01 µM- 100 µM]. **(C)** Inhibition of Ang-(1-7) formation by PRCP (10 ng/mL) was tested for five different inhibitors [0.01 µM- 100 µM]. **(D)** For PRCPi, lower inhibitor concentrations [0.01 nM- 100 nM] were used to investigate inhibitory effects on Ang-(1-7) formation by PRCP.

whole cell lysates is mainly driven by PRCP. In summary, in cell lysates, Ang-(1–7) formation by ACE2 was low, and most of the conversion was mediated by PRCP. Increased Ang-(1–7) formation rate upon FFSS cannot be attributed to ACE2 and PRCP, but rather indicates that another enzyme is involved.

**Cell supernatant.** To determine the contribution of ACE2 and PRCP in Ang-(1–7) formation in cell supernatants, hPC were incubated with and without ACE2i and PRCPi. Under no flow control conditions, ACE2 inhibition (mean 3.75 nmol/L/h) reduced the Ang-(1–7) formation rate significantly compared to total Ang-(1–7) formation rate (mean 8.02 nmol/L/h) and compared to PRCP inhibition (mean 7.70 nmol/L/h) (Fig 6A). Upon FFSS, Ang-(1–7) formation rate with PRCP inhibition (mean 5.55 nmol/L/h) was no longer significantly different from Ang-(1–7) formation rate with ACE2 inhibition (mean 3.54 nmol/L/h) (Fig 6B). This suggests, besides ACE2 being the main enzyme responsible for Ang-(1–7) formation in supernatants, PRCP-derived Ang-(1–7) formation was enhanced upon FFSS (Fig 6B). Taken together, Ang-(1–7) formation rate in supernatants is predominantly driven by ACE2 but contribution of PRCP increases upon FFSS suggesting increased enzyme transport to the cell surface through exocytosis.

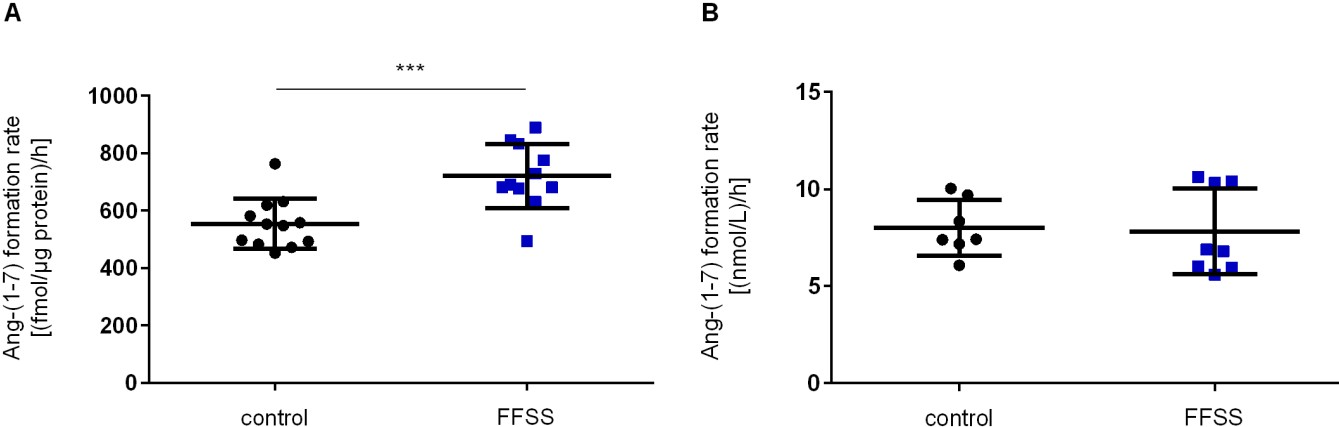

**Fig 3. Ang-(1-7) formation in hPC lysates and supernatants upon FFSS. (A)** Ang-(1-7) formation rate in cell lysates was elevated upon FFSS (blue squares) compared to no flow controls (black circles). Experiments were performed twice on different cell passages and different dates, each consisting of n = 6 technical replicates, respectively. Statistics: ***, p < 0.001, assessed by two-tailed Student´s t-test. **(B)** Ang-(1-7) formation rate in supernatants was not significantly regulated upon FFSS (blue squares) compared to no flow controls (black circles). Experiments were performed twice on different cell passages and different dates, each consisting of n = 4 technical replicates, respectively. Statistics: assessed by Mann-Whitney test. Each data point represents one technical replicate. Data are plotted as mean ± SD (horizontal lines).

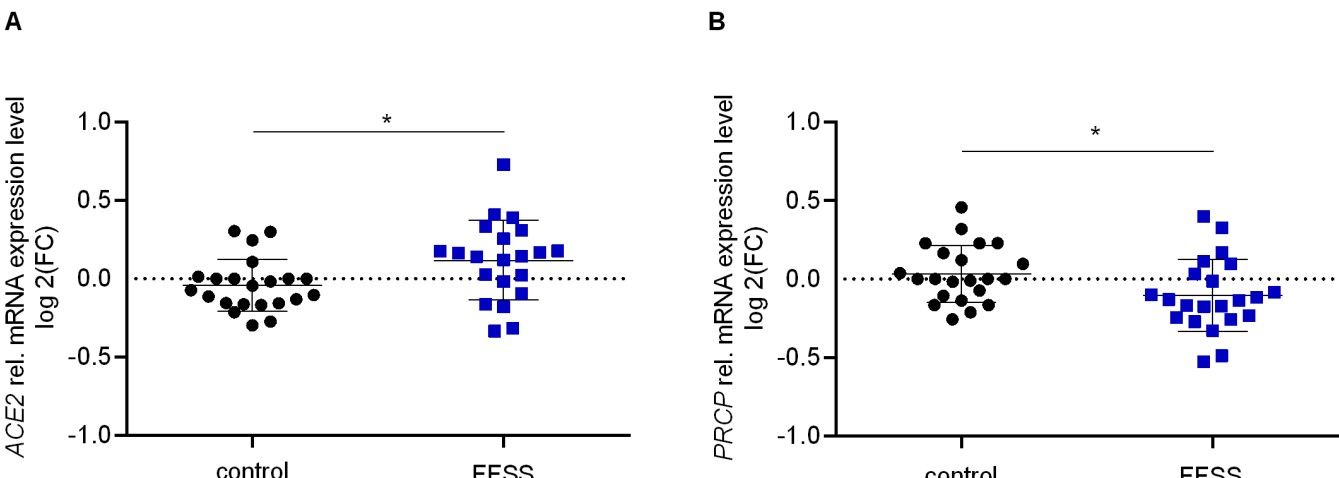

**Fig 4. Gene expression of ACE2 and PRCP in hPC upon FFSS. (A)** *ACE2* gene expression increased in hPC upon FFSS (blue squares) compared to no flow control (black circles). **(B)** *PRCP* gene expression decreased in hPC upon FFSS (blue squares) compared to no flow control (black circles). qPCR results are presented as relative mRNA expression level normalized to *GUSB* and referred to unstimulated control group. Each data point represents one technical replicate of four separate experiments. Experiments were performed on different cell passages and different dates, each experiment consisting of n = 5-6 technical replicates, respectively. Data are plotted as mean ± SD (horizontal lines). Statistics: *, *p* < 0.05, assessed by two-tailed Student´s t-test.

## Discussion

The intraglomerular RAS is implicated in the pathogenesis of progressive glomerular diseases, with podocyte injury playing a central role in glomerular damage and proteinuria. Renal pathologies are often associated with RAS activation and elevated levels of its primary effector, the vasoconstrictor Ang II [60–62]. The effects of Ang II on podocyte injury have been extensively studied and are associated with pathomechanisms underlying the progression of renal disease and

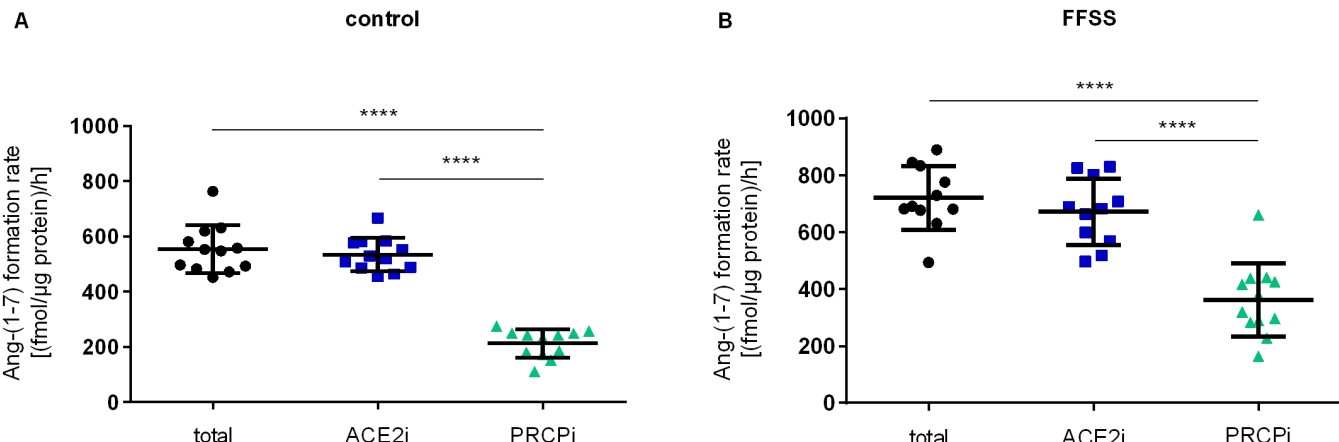

**Fig 5. Ang-(1-7) formation rate in hPC lysates upon FFSS. (A)** Ang-(1-7) formation rate under no flow control condition and **(B)** after FFSS exposure. Inhibition of ACE2 (blue squares) did not affect the Ang-(1-7) formation rate in hPC cell lysates compared to the total Ang-(1-7) formation rate (black circles). Inhibition of PRCP (green triangles) significantly reduced the Ang-(1-7) formation compared to total Ang-(1-7) formation (black circles) as well as ACE2 inhibition (blue squares). Experiments were performed twice on different cell passages and different dates, each experiment consisting of n = 6 technical replicates, respectively. Each data point represents one technical replicate. Data are plotted as mean ± SD (horizontal lines). Statistics: ****, p < 0.0001, assessed by one-way ANOVA with Tukey`s multiple comparison follow-up test.

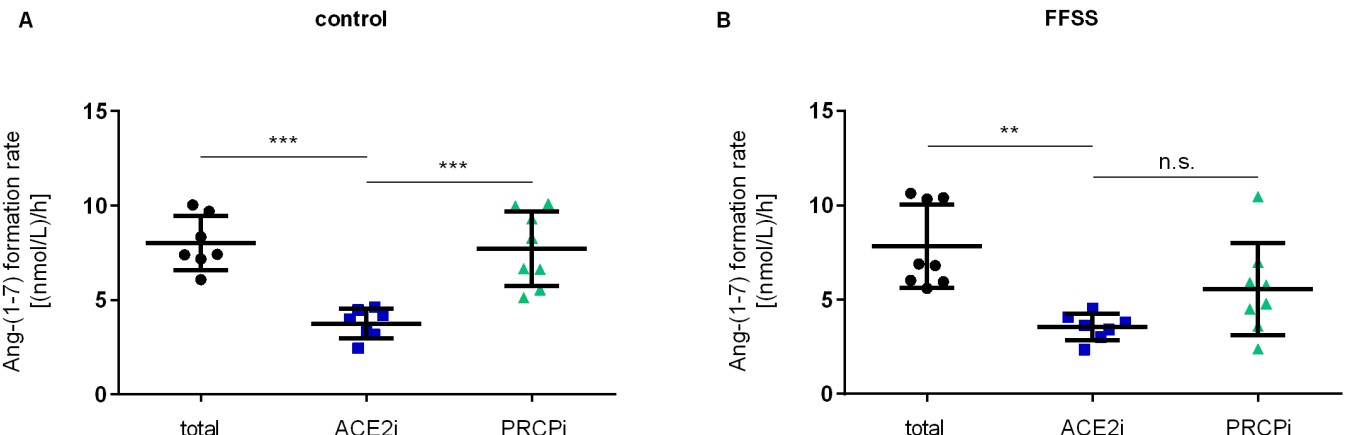

**Fig 6. Ang-(1-7) formation rate from Ang II in hPC supernatants upon FFSS. (A)** Under no flow control condition, Ang-(1-7) formation with ACE2 inhibition (blue squares) was significantly lower than total Ang-(1-7) formation (black circles) and formation with PRCP inhibition (green triangles). **(B)** Upon FFSS exposure, ACE2 inhibition (blue squares) reduced the Ang-(1-7) formation significantly compared to total Ang-(1-7) formation without inhibitors (black circles). Ang-(1-7) formation with PRCP inhibition (green triangles) was no longer significantly different to Ang-(1-7) formation upon ACE2 inhibition. Experiments were performed twice on different cell passages and different dates, each experiment consisting of n = 4 technical replicates, respectively. Statistics: **, p < 0.01; ***, p < 0.001; n.s., not significant, assessed by one-way ANOVA with Tukey`s multiple comparison follow-up test. Each data point represents one technical replicate. Data are plotted as mean ± SD (horizontal lines).

hypertension (reviewed in [41]). In hPC, Ang II induces the production of ROS and promotes cellular apoptosis via the Arf6-Erk1/2-Nox4 signaling pathway [38]. The molecular mechanisms involved in Ang II-induced podocyte apoptosis are mediated by AGT1R and include the activation of endoplasmic reticulum stress, protein kinase C delta, and the p38 MAPK pathway [36]. In immortalized mouse podocytes, mechanical strain triggered the activation of local tissue RAS, leading to elevated Ang II levels and AT1R-mediated podocyte apoptosis [31].

Our initial characterization of RAS components in differentiated hPC revealed gene expression of *ACE*, *ACE2*, *PRCP*, *PREP*, *AGTR1*, and weak expression of *AGTR2*. However, we did not detect expression of *AGT* and *REN*, despite previous reports indicating that podocytes possess an intrinsic and local RAS, which enables them to produce Ang II endogenously [31,63]. Notably, the expression of RAS components in podocytes has been investigated in several studies, with results varying significantly, suggesting that the experimental models and conditions used strongly influence the findings. For instance, in mice, podocytes exhibited ACE2 on protein level, while ACE was exclusively detected in glomerular endothelial cells [64]. In contrast, Velez et al. demonstrated functional ACE activity in cultured immortalized mouse podocytes and provided evidence for a functional intrinsic podocyte RAS [65].

However, the primary source of renal renin production remains localized to juxtaglomerular cells [66]. By combining tissue-specific KO models with a model of inducible podocyte injury, Matsusaka et al. showed that increased renal Ang II in response to podocyte injury was due to the increased filtration of systemic AGT, resulting from the loss of glomerular barrier function [67,68]. According to them, hepatic AGT is filtered through the glomerular filtration barrier and then further converted intrarenally.

In the present study, we characterized the enzymes involved in conversion of Ang II to Ang-(1–7) in hPC upon FFSS, using pharmacological inhibition of the potentially involved converting enzymes and LC/MS-MS analysis. FFSS parameters to model pathological hyperfiltration stress were used based on previous research. FFSS at the podocyte surface in healthy adult mice has been estimated to be approximately 0.3 dyne/cm² [24]. In models of unilateral nephrectomy reflecting a hyperfiltrating state, FFSS on mouse podocytes increases by 1.5–2.5-fold, reaching ~0.5–0.75 dyne/cm² [21]. As reported by others, applying 0.5 dyne/cm² on human primary podocytes promoted expression of podocyte-specific markers, supporting a physiological relevance of this stress level in hPC [27]. Based on these data, we selected 1 dyne/cm² for 2 hours to model pathological hyperfiltration, representing a mild but sustained increase over normal shear stress, sufficient to induce mechanosensitive responses in glomerular cells without causing acute cell detachment and loss.

Starting by testing the specificity of several inhibitors revealed MLN-4760 as a highly specific ACE2 inhibitor with an $IC_{50}$ of 508 nM. The dual PRCP and PREP inhibitor ZPP showed a higher efficacy for inhibiting PRCP with $IC_{50}$ values of 228 nM and 5.3 µM, respectively. PRCPi was highly selective for inhibiting PRCP ($IC_{50} = 78$ pM). The PREP inhibitor KYP-2047 also displayed an inhibitory effect on PRCP activity with an $IC_{50}$ of 8.3 µM and thus should not be considered a selective PREP inhibitor, especially when high inhibitor concentrations are used in the experimental setting. Our observations are consistent with those of Jalkanen et al., who previously reported a 20% inhibition of PRCP activity by 10 µM KYP-2047 in an in vitro pharmacological profiling study [69].

In an in vitro assay using pure recombinant human enzymes, we investigated the conversion of Ang II to Ang-(1–7) and detected Ang-(1–7) formation by ACE2 and PRCP. Although some studies reported PREP to convert Ang II to Ang-(1–7) [12,45,46], PREP did not catalyze this conversion into Ang-(1–7) in our setting. This is in line with findings by others that PREP primarily converts Ang I, and not Ang II, directly to Ang-(1–7) [47–49,54]. We confirmed this by using Ang I as a substrate and detected PREP-catalyzed Ang-(1–7) formation, thereby verifying the activity of our recombinant PREP enzyme (S5 Fig). It is noteworthy that in some studies ZPP, a dual PREP and PRCP inhibitor, was chosen. This complicates the interpretation of the reported results as it remains unclear which of the enzymes contributes to the observed effects and to which extend.

ACE2 has been identified as the main enzyme responsible for the conversion of Ang II to Ang-(1–7) in human kidney homogenates [49] and is thus often regarded as the primary enzyme responsible for directing the intrarenal RAS towards the vasodilatory and renoprotective axis by increasing Ang-(1–7) levels. ACE2 deficiency exacerbates kidney injury, while several kidney disease models show reduced renal ACE2 levels, as previously summarized in [17].

In the present study, we assessed the metabolism of Ang II to Ang-(1–7) in hPC cell lysates and on the cell surface, and evaluated the respective contributions of ACE2 and PRCP. On the cell surface, ACE2 activity was found to play a more prominent role in Ang-(1–7) formation than PRCP, especially under control conditions. This is consistent with ACE2

being predominantly localized at the plasma membrane, where it cleaves Ang II in the ultrafiltrate within Bowman's space [64,70–73]. In hPC lysates, Ang-(1–7) formation was predominantly driven by PRCP, reflecting both its higher expression levels compared to ACE2 and its mainly lysosomal localization, which enables it to cleave and degrade peptides intracellularly [74–76]. Upon FFSS, total Ang-(1–7) formation in supernatants was not altered but the contribution of PRCP in Ang-(1–7) formation was enhanced. This might result from increased enzyme transport to the cell surface through exocytosis or increased enzyme activity due to post-translational modification. Posttranslational modifications of PRCP were reported to include disulfide bonds, N-glycosylation sites, and both a signal sequence and a pro-peptide sequence, which may be cleaved upon maturation or activation [77,78].

In contrast to supernatants, our results reveal an increased conversion of Ang II to Ang-(1–7) in hPC lysates exposed to FFSS which was primarily driven by PRCP. The remaining Ang-(1–7) formation rate despite ACE2 and PRCP inhibition is suggestive of an additional enzyme involved in the conversion of Ang II in hPC lysates contributing to the increased Ang-(1–7) production observed upon FFSS. Additional enzymes involved in the degradation of angiotensin peptides that should be further investigated include Cathepsin A (CTSA; EC 3.4.16.5) and thimet oligopeptidase (THOP1; EC 3.4.24.15). Although CTSA is mainly known to hydrolyze Ang I [79], early studies suggest it may also cleave Ang II to Ang-(1–7) [80]. THOP1 is well established in cleaving Ang I to Ang-(1–7) [81,82], and further studies have indicated that it can also cleave Ang II, albeit inefficiently due to its low catalytic efficiency [83]. However, potential Ang II–derived cleavage products of THOP1 have not yet been identified. Future research might therefore specifically examine THOP1 and Cathepsin A under the experimental conditions in hPC to clarify their potential roles in local RAS regulation and defining more precisely the enzymatic mechanisms underlying Ang II processing in hPC.

In our study, PRCP is the predominant enzymatic contributor to Ang-(1–7) generation in podocyte lysates. PRCP cleaves carboxy-terminal residues from peptides with a penultimate proline, including Ang II and Ang III [78,84–86]. Its relevance for renal homeostasis is supported by findings that renal PRCP expression is reduced in hypertensive rats and restored by losartan [87], and that PRCP depletion in mice induced endothelial dysfunction, hypertension, alterations in renal and vascular eNOS, and increased ROS formation [88]. While FFSS did not directly activate PRCP, this enzymatic capacity provides a plausible mechanism to shift the local balance from Ang II toward Ang-(1–7). This is particularly relevant because Ang-(1–7) is well known to oppose Ang II–mediated injury by exerting vasodilatory, antioxidative, antiproliferative, antifibrotic, and anti-inflammatory actions [17].

Mechanical stress, including FFSS, is known to disrupt podocyte structure and survival by inducing cytoskeletal remodeling, detachment, and apoptosis, partly through mechanotransduction pathways involving Akt–GSK3β–β-catenin and MAPK/ERK signaling [24,89,90]. It has also been reported, that mechanical stress activates the local RAS, enhancing Ang II production and its downstream MAPK, TGF-β, and PI3K cascades, thereby reducing nephrin expression and promoting apoptosis and cytoskeletal destabilization [31,43,91,92]. Given these well-established Ang II-mediated injury mechanisms, a FFSS-associated shift toward Ang-(1–7) could be renoprotective and counterbalance FFSS-induced damage. Moreover, Ang-(1–7) has been demonstrated to directly signal protective in podocytes. It restored nephrin, podocin, and WT1 levels and reduced apoptosis under high-glucose conditions in mouse podocytes. These effects were blocked by the MAS1 receptor antagonist A-779 [93]. In human podocytes exposed to preeclampsia serum, Ang-(1–7) suppressed phosphorylation of p38, ERK1/2, and JNK in a MAS1 receptor-dependent manner, indicating inhibition of stress-activated MAPK pathways [94]. Ang-(1–7) also stimulated modest $AT_2R$-dependent nitric oxide production [95], a mediator of podocyte redox balance and cytoskeletal regulation [96]. The renoprotective effects of Ang-(1–7) extend to an in vivo mouse model, where cyclic Ang-(1–7) limited albuminuria progression, podocyte dysfunction, and glomerular fibrosis similarly to ACE inhibition [97].

The predominant contribution of PRCP to Ang-(1–7) formation supports the translational hypothesis that PRCP may shape how hPC engage Ang-(1–7)–dependent protective signaling under FFSS, thereby counteracting Ang II– and shear stress–induced injury.

It is well established that Ang-(1–7) signals through the MAS1 receptor, which we were able to detect in hPC via immunofluorescence (S6 Fig). Whether Ang-(1–7) acts in an autocrine manner on hPC or is directly degraded and inactivated remains to be determined. Another potential mechanism involves the internalization of the AT1R along with its ligand Ang II [13,17], followed by the intracellular conversion into Ang-(1–7). This process may exert additional protective effects through the reduction of Ang II levels. Further research is needed to determine if the proposed renoprotective role of PRCP can be functionally demonstrated, for instance by investigating whether PRCP inhibition exacerbates FFSS-induced damage, or whether PRCP overexpression or enhanced activity mitigates FFSS-induced injury.

As a limitation of our study, besides the in vitro setting of cell culture, we had to exogenously add Ang II, as the endogenous concentrations were too low to detect the cleavage pattern by the enzymes investigated. We deliberately chose a high Ang II concentration of 1 µg/mL to evaluate catalytic capacities under standardized conditions avoiding artificial limitations on enzyme activity. Although the exogenously applied Ang II concentration does not reflect physiological in vivo peptide levels, this approach enabled us to use spiked Ang II as a substrate, allowing clear identification of only the spiked product via mass spectrometry and the precise calculation of enzyme activity. Thus, this pharmacological approach contributes to a deeper understanding of Ang II to Ang-(1–7) conversion in human podocytes, particularly in the context of glomerular hyperfiltration. Yet, the molecular mechanisms and potential additional enzymes involved in upregulating Ang-(1–7) formation in response to FFSS need further investigations.

In conclusion, this study contributes to a better understanding of the intrarenal processing of Ang II towards the alternative RAS. Upon FFSS, Ang II to Ang-(1–7) conversion is elevated in hPC lysates which is mainly dependent on PRCP activity. Thus, PRCP may shape how hPC invoke Ang-(1–7)–dependent protective signaling under FFSS, thereby counteracting Ang II– and shear stress–induced injury.

## Supporting information

**S1 Table. Primer sequences for genes of interest.**
(PDF)

**S1 File. Supplement methods.**
(PDF)

**S1 Fig. Characterization of angiotensin system components on RNA level in differentiated hPC.**
(PDF)

**S2 Fig. Inhibitor effects on PREP-mediated Ang II to Ang-(1–7) conversion.**
(PDF)

**S3 Fig. Inhibition of Ang-(1–7) formation from substrate Ang II by different inhibitors.**
(PDF)

**S4 Fig. Western blot analyses of ACE2 and PRCP.**
(PDF)

**S5 Fig. Results NEP assay.**
(PDF)

**S6 Fig. Representative immunofluorescence image of MAS1.**
(PDF)

**S1 Data. Excel-file containing primary LC-MS/MS data.**
(XLSX)

**S2 Data. Excel-file calculation of delta delta Ct values for gene expression analyses of ACE2 and PRCP.**
(XLSX)

**S3 Data. Raw images.**
(PDF)

## Acknowledgments

We thankfully acknowledge the contributions of Karen Böhme and Petra Karsten for excellent laboratory assistance. We furthermore thank Moin A. Saleem, University of Bristol, UK for providing hPC, and the Sonnenfeld-Stiftung, Berlin, Germany for providing the FFSS device.

## Author contributions

**Conceptualization:** Debora Kaiser-Graf, Reinhold Kreutz, Juliane Bolbrinker.

**Data curation:** Debora Kaiser-Graf.

**Formal analysis:** Debora Kaiser-Graf, Juliane Bolbrinker.

**Investigation:** Debora Kaiser-Graf, Oliver Domenig.

**Methodology:** Debora Kaiser-Graf, Oliver Domenig, Marko Poglitsch, Juliane Bolbrinker.

**Project administration:** Reinhold Kreutz, Juliane Bolbrinker.

**Resources:** Oliver Domenig, Marko Poglitsch, Reinhold Kreutz.

**Software:** Debora Kaiser-Graf.

**Supervision:** Reinhold Kreutz, Juliane Bolbrinker.

**Validation:** Debora Kaiser-Graf.

**Visualization:** Debora Kaiser-Graf.

**Writing – original draft:** Debora Kaiser-Graf.

**Writing – review & editing:** Debora Kaiser-Graf, Oliver Domenig, Reinhold Kreutz, Juliane Bolbrinker.

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
