## [Decision Letter · Decision Letter 0]

22 Jul 2025

Dear Dr. Kaiser-Graf,

Thank you for submitting your manuscript to PLOS ONE. After careful consideration, we feel that it has merit but does not fully meet PLOS ONE’s publication criteria as it currently stands. Therefore, we invite you to submit a revised version of the manuscript that addresses the points raised during the review process.

We look forward to receiving your revised manuscript.

Kind regards,

Junzheng Yang

Academic Editor

PLOS ONE

Journal Requirements:

“I have read the journal's policy and the authors of this manuscript have the following competing interests: Oliver Domenig and Marko Poglitsch had a paid employment at Attoquant Diagnostics GmbH within the last 5 years.”

We note that one or more of the authors are employed by a commercial company: Attoquant Diagnostics GmbH

Reviewers' comments:

Reviewer's Responses to Questions

**Comments to the Author**

1. Is the manuscript technically sound, and do the data support the conclusions?

Reviewer #1: No

Reviewer #2: Yes

2. Has the statistical analysis been performed appropriately and rigorously?

Reviewer #1: No

Reviewer #2: Yes

3. Have the authors made all data underlying the findings in their manuscript fully available?

Reviewer #1: No

Reviewer #2: Yes

4. Is the manuscript presented in an intelligible fashion and written in standard English?

Reviewer #1: No

Reviewer #2: Yes

Reviewer #1: In this manuscript, authors investigated the conversion of Ang II to Ang-(1-7) in human podocytes (hPCs) both intracellularly and at the cell surface under fluid flow shear stress (FFSS). To do this, recombinant Ang II was added in hPC lysates (representing the intracellular state) or in the culture (supernatant) of the intact cells (representing the cell surface state). The converted products Ang-(1-7) were measured by means of LC-MS/MC. RT-qPCR and immunofluorescence were used to ascertain the presence of ACE2 and PRCP in hPCs. Authors concluded that more Ang-(1-7) was formed intracellularly under the FFSS treatment, but the conversion of Ang II to Ang-(1-7) stayed steady at the cell surface regardless of FFSS. With the aid of selected enzyme inhibitors, PRCP was found responsible for the Ang-(1-7) formation inside the cells while the conversion at the cell surface involved both ACE2 and PRCP enzymes. Authors also suggested that PRCP-derived Ang-(1-7) formation was enhanced upon FFSS in the supernatants of hPCs.

The rationale of using cell lysates to represent the intracellular state or the supernatant to represent the cell surface state was not explained. If cell lysates contain all intracellular proteins and cell surface proteins, and if added Ang II and the enzyme inhibitors were internalized into the intact hPCs during the one-hour incubation time, how would the result be interpreted? Where would be the major site of the conversion, inside the cells or on the cell surface?

The presentation and interpretation of the data were somewhat convoluted and hard to follow. Authors reported that the PRCP mRNA expression in hPCs was slightly decreased upon FFSS, yet the conversion of Ang II to Ang-(1-7) was increased intracellularly (cell lysate) under FFSS. This conversion was mainly driven by PRCP in cell lysates and enhanced at the cell surface (in supernatant) under FFSS. Please comment.

As authors indicated in the manuscript, Ang II was added into each of the testing samples as the substrate for its conversion to Ang-(1-7). Why would this in vitro data be extrapolatable to the in vivo setting of glomerular hyperfiltration?

Some specific concerns are listed below.

The data presented in the manuscript did not support the manuscript title which suggested a glomerular hyperfiltration setting.

Figure 1. The cell integrity is poor in general. Only one cell showed somewhat co-localization. The ACE2 image appeared different from the one in the merged image. It is not convincing if ACE2 is localized in the cytosol and/or on the cell surface membrane. Expressions of ACE2, PRCP, and PREP at the protein level require confirmation by other means.

Figure 2. The amount of Ang-(1-7) generated and detected was low in general throughout the entire study, given the amount of pure Ang II substrate added in the assays. How does this quantity relate to the physiological level of Ang-(1-7) in vivo? The number of hPCs, the amount of cell lysates, or the volume of supernatants used for each experiment was not clearly stated. This hampered the data interpretation. In some essays, the Ang-(1-7) formation rate was reported and in others the Ang-(1-7) formation was reported. What does the “rate” mean in this context? Please comment. Methods of LC-MS/MS were not described in detail.

Figure 3. The actual mRNA expression levels of ACE2 and PRCP were not reported, rather, the fold change between the control and the FFSS groups was reported, despite the neglectable difference of the expression level between sample groups.

Figure 4 and Figure 5. These were basically the same experimental design testing the formation of Ang-(1-7) in cell lysates (Figure 4) or in the supernatants (Figure 5). The combined inhibitors were used in Figure 4, but in Figure 5 single inhibitor was used. Please comment.

If intracellular Ang-(1-7) formation in hPCs was mainly driven by PRCP (Figure 4), following from this, would Figure 5 suggest that both ACE2 and PRCP drove the conversion of Ang II to Ang-(1-7) in the supernatant? Please comment.

If PRCP-derived Ang-(1-7) formation was enhanced upon FFSS in supernatants, similarly, would Figure 5 suggest that ACE2-derived Ang-(1-7) formation was reduced upon FFSS in supernatant? Please comment.

448-449, “on the cell surface, ACE2 activity was found to play a more prominent role in Ang-(1-7) formation than PRCP, especially under control conditions”. If this statement is true, why would ACE2 not be a major player in Ang-(1-7) formation in cell lysates. Please clarify.

Overall, without clear evidence of ACE2 and PRCP expression/presence in hPCs, the results of ACE2- or PRCP-derived Ang-(1-7) formation in cell lysates or supernatants remain incidental.

Reviewer #2: 1. Fig 1- Green channels of ACE2 and PREP seem to have been duplicated.

I think the green channel shown for ACE2 IF actually belongs to PREP IF. The authors must rectify this error.

2. Fig 1 - Figure numbers A, B and C are missing for this figure but are referenced in the text.

3. Fig 1 - Not all cells have a colocalization between PRCP and LAMP1.

4. Fig 1 - It is unclear how the authors conclude that there is an apparent difference in colocalization between PREP and PRCP and LAMP1 without any quantitative measure. To me, they look similarly partially colocalized. The authors should provide quantitative measures to support their conclusion and provide some more images of different regions and replicates in the Supplementary figures to generate confidence in their data.

5. The authors should provide the rationale why they investigated the colocalization of various enzymes and LAMP1, which will be useful for the general reader to understand this study.

6. The authors should make more efforts to elaborate a little on each of their main results and their implications. Otherwise, the reader has to wait until the Discussion section to gain insight into the current work. This will enhance the readability of the manuscript by making it more engaging.

7. Consider incorporating Fig. S2(A-D) part of the main figures because of their importance to understanding the results of this study.

**Do you want your identity to be public for this peer review?** For information about this choice, including consent withdrawal, please see our Privacy Policy

Reviewer #1: No

Reviewer #2: No

---

## [Author Response · Author response to Decision Letter 1]

4 Sep 2025

PONE-D-25-20067 - Revision (please see the word document "Responses to reviewers")

Response to Reviewers and Editors

Response to Editors

We thank the Editor for the careful evaluation of our manuscript. Your valuable comments have helped us to substantially improve the clarity and precision of our work.

As a general note, all page and line numbers in our responses refer to the revised and marked manuscript version titled “Revised Article with Changes Highlighted”, unless otherwise specified.

Response: We carefully checked the PLOS ONE’s style requirements and adjusted several headings to sentence case:

- Page 5, line 108: Cell culture

- Page 5, line 122: RNA isolation and reverse transcription

- Page 10, line 213: Quantitative real-time PCR

Additionally, we deleted the numbers from the subheadings of the Method section and we adjusted the naming of our Supporting Information files at the end of the revised manuscript in the section titled “Supporting Information” (page 42, line 955-975).

“I have read the journal's policy and the authors of this manuscript have the following competing interests: Oliver Domenig and Marko Poglitsch had a paid employment at Attoquant Diagnostics GmbH within the last 5 years.”

We note that one or more of the authors are employed by a commercial company: Attoquant Diagnostics GmbH

Response: As requested, our amended Funding Statement now reads:

“Oliver Domenig and Marko Poglitsch had a paid employment at Attoquant Diagnostics GmbH within the last 5 years. Attoquant Diagnostics GmbH provided support in the form of salaries for authors [OD, MP], but did not have any additional role in the study design, data collection and analysis, decision to publish, or preparation of the manuscript. The specific roles of these authors are articulated in the Author Contributions section.”

We reviewed our statements relating to the author contributions, and indicated the roles that these authors had in our study correctly. Thus, an update in the Author Contributions section of the online submission form is not necessary.

Response: As requested, our updated Competing Interests Statement now reads:

“Oliver Domenig and Marko Poglitsch had a paid employment at Attoquant Diagnostics GmbH within the last 5 years. This commercial affiliation does not alter our adherence to PLOS ONE policies on sharing data and materials.”

We included both the updated Funding Statement and Competing Interests Statement in our cover letter.

Response: We acknowledge this important note by the Editors and decided to include our entire data within the Supporting Information. Excel-file S1_Data contains primary LC-MS/MS data and S2_Data Excel-file calculation of delta delta Ct values for gene expression analyses of ACE2 and PRCP.

Response: Thank you for this comment. We added the information in the Supporting Information files as S5_Fig and S6_Fig. We updated the corresponding parts in the revised manuscript and deleted the phrase “data not shown”.

Response: We carefully checked the requirements for uncropped and unadjusted images underlaying blot and gel data as reported in the author guidelines. We have added the newly acquired western blot data to our raw data file. Raw images were annotated and compiled according to the author guidelines. Briefly, original images were annotated using GIMP and exported as tif-files. Tif-files were compiled into one pdf file using Adobe Acrobat. The blot/gel image data are in the Supporting Information “S1_raw_data” file. We added this information in our cover letter.

Response: Reviewers did not recommend to cite specific previously published works.

Response to Reviewer 1 Comments

Comment 1.1

In this manuscript, authors investigated the conversion of Ang II to Ang-(1-7) in human podocytes (hPCs) both intracellularly and at the cell surface under fluid flow shear stress (FFSS). To do this, recombinant Ang II was added in hPC lysates (representing the intracellular state) or in the culture (supernatant) of the intact cells (representing the cell surface state). The converted products Ang-(1-7) were measured by means of LC-MS/MC. RT-qPCR and immunofluorescence were used to ascertain the presence of ACE2 and PRCP in hPCs. Authors concluded that more Ang-(1-7) was formed intracellularly under the FFSS treatment, but the conversion of Ang II to Ang-(1-7) stayed steady at the cell surface regardless of FFSS. With the aid of selected enzyme inhibitors, PRCP was found responsible for the Ang-(1-7) formation inside the cells while the conversion at the cell surface involved both ACE2 and PRCP enzymes. Authors also suggested that PRCP-derived Ang-(1-7) formation was enhanced upon FFSS in the supernatants of hPCs.

The rationale of using cell lysates to represent the intracellular state or the supernatant to represent the cell surface state was not explained. If cell lysates contain all intracellular proteins and cell surface proteins, and if added Ang II and the enzyme inhibitors were internalized into the intact hPCs during the one-hour incubation time, how would the result be interpreted? Where would be the major site of the conversion, inside the cells or on the cell surface?

Response:

We thank the Reviewer for the careful evaluation of our manuscript. Your valuable comments have helped us to substantially improve the clarity and precision of our work.

As a general note, all page and line numbers in our responses refer to the revised and marked manuscript version titled “Revised Article with Changes Highlighted”, unless otherwise specified.

In addition, all raw data are now provided in the revised Supplementary Material.

Since whole-cell lysates contain both intracellular and membrane-associated proteins, we agree that the term “intracellular” was inaccurate in this context. We have carefully revised the manuscript and removed the term “intracellular” from page 9, line 223; page 13, line 281; page 15, lines 339 and 347; and page 26, line 386 of the original version. To improve accuracy, we now explicitly state “in cell lysates” where appropriate. For example, on page 16, lines 379–381, the text now reads:

“In contrast to elevated Ang-(1-7) formation in cell lysates, no effect was observed on Ang-(1-7) formation rate at the cell surface of hPC (Fig. 3B).”

Similarly, the revised text on page 19, lines 447–448, reads:

“However, Ang-(1-7) formation in hPC whole-cell lysates is mainly driven by PRCP.”

In the Discussion section (page 22, lines 502–503), we revised the sentence to:

“Our results revealed an increased conversion of Ang II to Ang-(1-7) in hPC lysates exposed to FFSS.”

For the measurements of Ang-(1–7) formation in whole-cell lysates, cells were sonicated before the addition of spiked Ang II and enzyme inhibitors. This procedure was described in the original Methods section 7.1 (page 10, lines 220–224):

“Cell suspensions were sonicated for complete dissolution using Vibra-Cell™ (Sonics & Materials Inc., Newtown, CT, USA) and normalized to total protein concentration determined using the BIO-RAD Protein Assay Dye with bovine serum albumin as protein standard (cat. no. 5000006, Hercules, CA, USA).”

Therefore, internalization of Ang II is not relevant in this context.

In contrast, for extracellular Ang-(1–7) formation in supernatants, Ang II and enzyme inhibitors were added to intact cells. During the one-hour incubation period, receptor-mediated internalization of Ang II could have occurred. However, such potential internalization would have affected both experimental conditions (control and FFSS) equally. Importantly, Ang II was added in saturating concentrations, and residual substrate levels were measured after incubation to confirm that enzyme activities were not substrate-limited. In addition, there is, to our knowledge, no rationale for enzyme inhibitors to undergo internalization in intact cells.

Regarding the Reviewer’s question about the major site of Ang II conversion to Ang-(1-7), we consider it most likely to occur inside the cells. This is supported by the low contribution of ACE2 to Ang-(1-7) formation in whole-cell lysates and the presumably higher intracellular content of PRCP. A direct quantitative comparison between lysates and supernatants is not feasible, however, since Ang-(1-7) formation in lysates was normalized to protein content, whereas absolute values were determined in supernatants.

Comment 1.2

The presentation and interpretation of the data were somewhat convoluted and hard to follow.

Response: We thank the reviewer for this overall comment. We acknowledge, that the flow of the manuscript may have been difficult to follow for the reader. To improve clarity and readability, we thoroughly revised the manuscript and expanded on the main findings and their implications within the Results section (see also Reviewer 2, comment 6).

To enhance data presentation and to facilitate interpretation, we have incorporated Supplementary Figure S2A-D into the main text of the revised manuscript, now presented as Figure 2, in line with the recommendation of Reviewer 2.

In addition, we restructured the manuscript to provide a more logical sequence of methods and results. Specifically, the methodology and results of the inhibitor testing were moved forward to directly follow enzyme characterization. This section now appears as Methods section 4 (“Inhibition of Ang-(1-7) formation by different inhibitors,” page 7, line 163 ff.) and in the revised Results section as “Enzymes involved in Ang-(1-7) formation from substrate Ang II” (page 15, line 339 ff.).

Finally, the presentation of the data in the original Figures 4 and 5 (now revised Figures 5 and 6) has been improved for greater clarity. For further details on these revisions, please see our response to Comment 1.12.

Comment 1.3

Authors reported that the PRCP mRNA expression in hPCs was slightly decreased upon FFSS, yet the conversion of Ang II to Ang-(1-7) was increased intracellularly (cell lysate) under FFSS. This conversion was mainly driven by PRCP in cell lysates and enhanced at the cell surface (in supernatant) under FFSS. Please comment.

Response:

We thank the Reviewer for this insightful comment. In our measurements of PRCP mRNA expression in cell lysates, we observed a slight decrease under FFSS (revised Figure 4). In contrast, the total Ang-(1–7) formation rate was significantly increased under FFSS conditions (revised Figure 3). As the Reviewer correctly notes, Ang-(

---

## [Decision Letter · Decision Letter 1]

21 Oct 2025

Dear Dr. Kaiser-Graf,

Thank you for submitting your manuscript to PLOS ONE. After careful consideration, we feel that it has merit but does not fully meet PLOS ONE’s publication criteria as it currently stands. Therefore, we invite you to submit a revised version of the manuscript that addresses the points raised during the review process.

We look forward to receiving your revised manuscript.

Kind regards,

Junzheng Yang

Academic Editor

PLOS ONE

Journal Requirements:

Reviewers' comments:

Reviewer's Responses to Questions

**Comments to the Author**

Reviewer #2: All comments have been addressed

Reviewer #3: (No Response)

Reviewer #4: (No Response)

2. Is the manuscript technically sound, and do the data support the conclusions?

Reviewer #2: Yes

Reviewer #3: Partly

Reviewer #4: Yes

3. Has the statistical analysis been performed appropriately and rigorously?

Reviewer #2: Yes

Reviewer #3: Yes

Reviewer #4: Yes

4. Have the authors made all data underlying the findings in their manuscript fully available?

Reviewer #2: Yes

Reviewer #3: Yes

Reviewer #4: Yes

5. Is the manuscript presented in an intelligible fashion and written in standard English?

Reviewer #2: Yes

Reviewer #3: Yes

Reviewer #4: Yes

Reviewer #2: (No Response)

Reviewer #3: Reviewer 1

Due to limitations in study design some comments could not be responded succesfully.

Comment 1.1

This comment has been partially addressed.

a. What was the composition of the buffer used for the preparation ? did it have any detergents? ionic strenght?

b. Cells suspensions were sonicated for complet dissolution... I think complete dissolution of membranes is not possible with this methodology.

c. Was there any centrifugation step involved?

What was the duration and conditions of centrifugation? This is important to know what type of membranes author worked with

If no centrifugation was used the protocol seems not appropriate.

Comment 1.3

This comment has not been addressed. Authors postulate now the participation of another enzyme that may contribute to the increased Ang-(1-7) formation observed under FFSS in cell lysastes

Which enzyme do authors considered ??

Comment 1.4

I concur with Reviewer 1 int that Ang II levels used were too high to make an extrapolation to in vivo settings

Comment 1.15

ACE2 activity coming from the cell surface cannnot be evaluated with this protocol since all membranes were mix together after sonication and centrifugation of lysates

Reviewer #4: This study clarifies how fluid shear stress (FFSS) regulates Angiotensin-(1-7) formation in human podocytes. The authors show that the intracellular enzyme PRCP, not ACE2, is the dominant source of Ang-(1-7) production, a potentially protective pathway enhanced by FFSS. Here, a few concerns regarding data interpretation and discussion need to be addressed.

1. The residual Ang-(1-7) formation in hPC lysates after dual ACE2 and PRCP inhibition (Fig 5) is noted to be upregulated by FFSS. This finding is a key mechanistic gap, the authors hypothesize another enzyme is involved but offer no data. Experimental or literature-based discussion is essential to hypothesize which other peptidases (e.g., Cathepsin A) could account for this activity, thereby providing a complete map of the RAS balance in the podocyte and guiding future research.

2. This study's conclusion rests on the idea that the PRCP-driven Ang-(1-7) increase is a "renoprotective" shift in the RAS balance, but this is never demonstrated functionally in the model. Determine if PRCP inhibition significantly exacerbates this FFSS-induced injury, thereby validating its protective role.

3. The authors noted the discrepancy: PRCP mRNA was slightly reduced while total Ang-(1-7) formation (driven by PRCP in lysate) was increased and hypothesized this was due to post-translational modifications. Perform quantitative densitometry on the existing Western Blot data (S4 Fig) for ACE2 and PRCP to confirm that the observed changes in enzyme activity are independent of changes in total protein mass.

4. They established the significance of Ang-(1-7) from the literature but did not functionally validate that the PRCP-driven increase specifically confers protection in their FFSS model. Adapt the Discussion to explicitly link the observed PRCP activation under FFSS with the known molecular protective signaling of Ang-(1-7), framing this as the key translational hypothesis.

5. Briefly justify the physiological relevance of the 1 dyne/cm2 for 2 h FFSS parameters in the Discussion to confirm they model pathological hyperfiltration stress.

6. The authors observed low ACE2 contribution in the cell lysates but high ACE2 contribution in the supernatant. Revise the Discussion and explicitly clarify how the differential compartmentalization (membrane vs. intracellular) accounts for the highly distinct enzymatic roles of ACE2 and PRCP across the two assay types.

**Do you want your identity to be public for this peer review?** For information about this choice, including consent withdrawal, please see our Privacy Policy

Reviewer #2: No

Reviewer #3: No

Reviewer #4: **Yes: ** Anjan K Bongoni

---

## [Author Response · Author response to Decision Letter 2]

4 Dec 2025

PONE-D-25-20067R1 – Revision

Response to Reviewers and Editors

Response to Editors

We thank the Editor for the thorough evaluation of our manuscript. In the following, we address the general remarks provided:

Journal Requirements:

Response: Thank you for the advice. However, the Reviewers did not propose any specific references.

Response: We carefully reviewed our reference list and checked for retracted articles. We did not find any retracted article in our list. However, reference number 65 (Velez JC, Bland AM, Arthur JM, Raymond JR, Janech MG. Characterization of renin-angiotensin system enzyme activities in cultured mouse podocytes. Am J Physiol Renal Physiol. 2007;293(1): F398–407. Epub 20070411. doi: 10.1152/ajprenal.00050.2007. PubMed PMID: 17429035.) was corrected in 2008, as the authors had mistakenly labeled a measured peptide as Ang-(1-9), which was later found to be Ang-(2-10) [1]. The correction does not affect our citation of the study. Therefore, we have retained this reference.

We also identified an updated PMID number for reference number 2 and updated it in our reference list (Inagami T. The renin-angiotensin system. Essays Biochem. 1994; 28:147–64. PubMed PMID: 7925317).

Response to Review Comments to the Author

Response to Reviewer #2: (No Response)

Response: We are glad that we were able to successfully address all of Reviewer’s #2 questions and comments.

Response to Reviewer #3:

Due to limitations in study design some comments could not be responded successfully.

Response: We thank the Reviewer #3 for carefully reading our manuscript and evaluating our responses to the Reviewers #1 and #2. Your comments helped us further improve our manuscript.

As a general note, all page and line numbers in our responses refer to the latest revised and marked manuscript version titled “Revised Article with Changes Highlighted”, unless otherwise specified.

Comment 1.1

This comment has been partially addressed.

Response: We thank the author for this comment and noticed that we need to explain the methodology in more detail.

a. What was the composition of the buffer used for the preparation? did it have any detergents? ionic strength?

Response: We used PBS (pH 7.4) to assess Ang-(1-7) formation in hPC cell lysates. Cells were scraped from the slides and transferred to PBS prior to sonification as described in the original manuscript on page 9 lines 220-221:” Cells were scratched from slides and transferred to a vial containing PBS.” We added the product information when first mentioned in the manuscript Method section – immunofluorescence in lines 133-134, where it now reads: “Medium was discarded and cells washed twice with phosphate buffered saline (PBS, cat. no. 14190094, Thermo Fisher Scientific, Waltham, MA, US).”

No detergents were used in the sample preparation procedure of cell lysates. To calculate the ionic strength, we relied on the manufacturer's specifications for the salt composition of PBS, assumed an ideal diluted solution and that all phosphorous compounds are dissociated completely. The resulting ionic strength for PBS at pH 7.4 is 166 mM.

b. Cells suspensions were sonicated for complete dissolution... I think complete dissolution of membranes is not possible with this methodology.

Response: We thank the reviewer for this comment. Cells were lysed using Vibra-Cell™ and membranes were destroyed by sonication using this methodology. We agree with the Reviewer, that the word “complete” cannot be used with absolute certainty in this context and therefore, deleted it from the sentence. The revised manuscript now reads in lines 221-223: “Cell suspensions were sonicated for dissolution using Vibra-Cell™ (Sonics & Materials Inc., Newtown, CT; US) and centrifugated at 16260 rcf for 10 min at 4°C.”

c. Was there any centrifugation step involved?

What was the duration and conditions of centrifugation? This is important to know what type of membranes author worked with. If no centrifugation was used the protocol seems not appropriate.

Response: That is a valuable comment. Indeed, we performed a centrifugation step. We sincerely apologize for our inaccuracy in the description of the methodology. For the cell lysates sample preparation, samples were centrifuged at 4°C for 10 minutes and 16260 rcf after sonication. Supernatants were collected and protein concentration quantified. We have amended the methodology in the revised manuscript. It now reads in lines 221-226: “Cell suspensions were sonicated for dissolution using Vibra-Cell™ (Sonics & Materials Inc., Newtown, CT; US) and centrifugated at 16260 rcf for 10 min at 4°C. Supernatants were collected and normalized to total protein concentration determined using BIO-RAD Protein Assay Dye with bovine serum albumin as protein standard (cat. no. 5000006, Hercules, CA, US).”

Comment 1.3

This comment has not been addressed. Authors postulate now the participation of another enzyme that may contribute to the increased Ang-(1-7) formation observed under FFSS in cell lysates. Which enzyme do authors considered?

Response: We recognize that further consideration and clarification are needed regarding additional potential enzymes involved in Ang-(1-7) formation, particularly as Reviewer #4 has raised a similar concern. We agree that at least a literature-based hypothesis, as suggested by Reviewer #4, is necessary to address this mechanistic gap. Thus, we conducted an extensive literature search to identify enzymes that could potentially account for the observed conversion of Ang II to Ang-(1-7) besides ACE2, PRCP and PREP:

Cathepsin A (CTSA), a lysosomal enzyme widely expressed in human tissues including the kidney [2], was initially proposed as a candidate for Ang-(1-7) formation. Early studies by Grynbaum and Marks reported that Cathepsin A from rat brain could release the C-terminal Phe from Ile^5-Ang II, producing Ang-(1-7) [3]. In contrast, Miller et al. found that Cathepsin A from porcine kidney cleaved Ang I to Ang-(1-9) and to Ang II, but did not hydrolyze Ang II [4]. Subsequent studies similarly highlighted Ang I as a primary substrate of Cathepsin A [5].

Grobe et al. reported that nephrolysin (NEP), thimet oligopeptidase (THOP1), and neurolysin (NLN) can generate Ang-(1-7), though they cleave Ang I and not Ang II [6]. Both NLN and THOP1 cleave Ang I at the Pro-Phe bond to form Ang-(1-7) [7, 8]. NLN is ubiquitously expressed, including in the kidney [9], but is primarily studied in the brain, where it is associated with stroke [10, 11]. While NLN can inactivate Ang II [8, 11, 12], this produces Ang-(1–4), not Ang-(1–7) [13]. Consequently, it is not a potential enzyme for our investigated conversion of Ang II to Ang-(1-7).

Velez et al. suggested that THOP1 may also convert Ang II to Ang-(1-7), alongside ACE2, RPCP, and PREP [14]. Early assays found no Ang II cleavage by recombinant rat THOP1 [15], while later studies suggested inefficient cleavage due to low catalytic efficiency [16]. Since existing statements are contradictory, we conclude that it remains unclear whether THOP1 actually hydrolyzes Ang II besides Ang I. Furthermore, no studies have identified specific Ang II–derived products of THOP1. Thus, THOP1 might inactivate Ang II by generating Ang-(1–4) or it might produce Ang-(1–7).

Overall, current evidence does not conclusively show that any of these enzymes convert Ang II to Ang-(1–7). Further studies are needed to define the enzymes regulating the RAS in podocytes, particularly experimentally assessing THOP1 and Cathepsin A under our conditions in hPC. We addressed these considerations in the revised Discussion section, which now reads in lines 522-533: “Additional enzymes involved in the degradation of angiotensin peptides that should be further investigated include Cathepsin A (CTSA; EC 3.4.16.5) and thimet oligopeptidase (THOP1; EC 3.4.24.15). Although CTSA is mainly known to hydrolyze Ang I [79], early studies suggest it may also cleave Ang II to Ang-(1–7) [80]. THOP1 is well established in cleaving Ang I to Ang-(1–7) [81,82], and further studies have indicated that it can also cleave Ang II, albeit inefficiently due to its low catalytic efficiency [83]. However, potential Ang II–derived cleavage products of THOP1 have not yet been identified. Future research might therefore specifically examine THOP1 and Cathepsin A under the experimental conditions in hPC to clarify their potential roles in local RAS regulation and defining more precisely the enzymatic mechanisms underlying Ang II processing in hPC.”

Comment 1.4

I concur with Reviewer 1 int that Ang II levels used were too high to make an extrapolation to in vivo settings

Response: We thank the Reviewer for raising this concern once again. As noted in our response to Comment 1.4 in our first revision, we agreed and acknowledged that the concentration of Ang II used in our experiments does not reflect in vivo peptide levels and represents a limitation of the study. To more clearly emphasize the importance of this comment, we have highlighted this limitation explicitly in the discussion section of the newly revised manuscript. It now reads in lines 604-610: “We deliberately chose a high Ang II concentration of 1µg/mL to evaluate catalytic capacities under standardized conditions avoiding artificial limitations on enzyme activity. Although the exogenously applied Ang II concentration does not reflect physiological in vivo peptide levels, this approach enabled us to use spiked Ang II as a substrate, allowing clear identification of only the spiked product via mass spectrometry and the precise calculation of enzyme activity.”

Comment 1.15

ACE2 activity coming from the cell surface cannot be evaluated with this protocol since all membranes were mix together after sonication and centrifugation of lysates

Response: Two different experimental setups were used to assess enzyme activity in cell lysates and on the cell surface. For cell lysate measurements, cells were sonicated as described in the Method section “Cell lysates sample preparation”. Enzyme activities at the cell surface were determined using intact cells as described in the Method section “Cell supernatants sample preparation”, where Ang-(1-7) formation was measured in supernatants of intact cells. To highlight the use of cell supernatants we wrote in the revised manuscript in lines 238-241: “100µL inhibitor solutions were applied each into one chamber of the cassette onto hPC and incubated for 1 h at 37 °C. Supernatants were collected and, following stabilization by acidification, Ang-(1-7) formation was quantified by LC-MS/MS (Methods 7.3).”

Response to Reviewer #4:

This study clarifies how fluid shear stress (FFSS) regulates Angiotensin-(1-7) formation in human podocytes. The authors show that the intracellular enzyme PRCP, not ACE2, is the dominant source of Ang-(1-7) production, a potentially protective pathway enhanced by FFSS. Here, a few concerns regarding data interpretation and discussion need to be addressed.

Response: We appreciate the Reviewer’s overall feedback, as well as his thorough and thoughtful comments provided below. They have been extremely valuable and we hope that we have addressed them sufficiently.

As a general note, all page and line numbers in our responses refer to the latest revised and marked manuscript version titled “Revised Article with Changes Highlighted”, unless otherwise specified.

1. The residual Ang-(1-7) formation in hPC lysates after dual ACE2 and PRCP inhibition (Fig 5) is noted to be upregulated by FFSS. This finding is a key mechanistic gap, the authors hypothesize another enzyme is involved but offer no data. Experimental or literature-based discussion is essential to hypothesize which other peptidases (e.g., Cathepsin A) could account for this activity, thereby providing a complete map of the RAS balance in the podocyte and guiding future research.

Response: We thank the Reviewer for this valuable comment and acknowledge the importance of further elaborating on additional peptidases involved in Ang-(1-7) formation. We conducted an extensive literature search to identify enzymes that might account for the observed conversion of Ang II to Ang-(1-7).

Cathepsin A (CTSA), a lysosomal enzyme widely expressed in human tissues including the kidney [2], was initially proposed as a candidate for Ang-(1-7) formation. Early studies by Grynbaum and Marks reported that Cathepsin A from rat brain could release the C-terminal Phe from Ile^5-Ang II, producing Ang-(1-7) [3]. In contrast, Miller et al. found that Cathepsin A from porcine kidney cleaved Ang I to Ang-(1-9) and to Ang II, but did not hydrolyze Ang II [4]. Subsequent studies similarly highlighted Ang I as a primary substrate of Cathepsin A [5].

Grobe et al. reported that nephrolysin (NEP), thimet oligopeptidase (THOP1), and neurolysin (NLN) can generate Ang-(1-7), though they cleave Ang I and not Ang II [6]. Both NLN and THOP1 cleave Ang I at the Pro-Phe bond to form Ang-(1-7) [7, 8]. NLN is ubiquitously expressed, including in the kidney [9], but is primarily studied in the brain, where it is associated with stroke [10, 11]. While NLN can inactivate Ang II [8, 11, 12], this produces Ang-(1–4), not Ang-(1–7) [13]. Consequently, it is not a potential enzyme for our investigated conversion of Ang II to Ang-(1-7).

Velez et al. suggested that THOP1 may also convert Ang II to Ang-(1-7), alongside ACE2, RPCP, and PREP [14]. Early assays found no Ang II cleavage by recombinant rat THOP1 [15], while later studies suggested inefficient cleavage due to low catalytic efficiency [16]. Since existing statements are contradictory, we conclude that it remains unclear whether THOP1 actually hydrolyzes Ang II besides Ang I. Furthermore, no studies have identified specific Ang II–derived products of THOP1. Thus, THOP1 might inactivate Ang II by generating Ang-(1–4) or it might produce Ang-(1–7).

Overall, current evidence does not conclusively show that any of these enzymes convert Ang II to Ang-(1–7). Further studies are needed to define the enzymes regulating the RAS in podocytes, particularly experimentally assessing THOP1 and Cathepsin A under our conditions in hPC. We addressed these considerations in the revised Discussion section, which now in lines 522-533: “Additional enzymes involved in the degradation of angiotensin peptides that should be further investigated include Cathepsin A (CTSA; EC 3.4.16.5) and thimet oligopeptidase (THOP1; EC 3.4.24.15). Although CTSA is mainly known to hydrolyze Ang I [79], early studies suggest it may also cleave Ang II to Ang-(1–7) [80]. THOP1 is well established in cleaving Ang I to Ang-(1–7) [81,82], and further studies have indicated that it can also cleave Ang II, albeit inefficiently due to its low catalytic efficiency [83]. However, potential Ang II–derived cleavage products of THOP1 have not yet been identified. Future research might therefore specifically examine THOP1 and Cathepsin A under the experimental conditions in hPC to clarify their potential roles in local RAS regulation and defining more precisely the enzymatic mechanisms underlying Ang II processing in hPC.”

2. This study's conclusion rests on the idea that the PRCP-driven Ang-(1-7) increase is a "renoprotective" shift in the RAS balance, but this is never demonstrated functionally in the model. Determine if PRCP inhibition significantly exacerbates this FFSS-induced inj

---

## [Decision Letter · Decision Letter 2]

14 Dec 2025

Upregulation of angiotensin-(1-7) formation in human podocytes – enzyme activity assay upon fluid flow shear stress

PONE-D-25-20067R2

Dear Dr. Kaiser-Graf,

We’re pleased to inform you that your manuscript has been judged scientifically suitable for publication and will be formally accepted for publication once it meets all outstanding technical requirements.

Kind regards,

Junzheng Yang

Academic Editor

PLOS One

Additional Editor Comments (optional):

Reviewers' comments:

Reviewer's Responses to Questions

**Comments to the Author**

Reviewer #4: All comments have been addressed

2. Is the manuscript technically sound, and do the data support the conclusions?

Reviewer #4: Yes

3. Has the statistical analysis been performed appropriately and rigorously?

Reviewer #4: Yes

4. Have the authors made all data underlying the findings in their manuscript fully available?

Reviewer #4: Yes

5. Is the manuscript presented in an intelligible fashion and written in standard English?

Reviewer #4: Yes

Reviewer #4: The authors have provided thorough, justified responses and made appropriate revisions to the discussion section.

**Do you want your identity to be public for this peer review?** For information about this choice, including consent withdrawal, please see our Privacy Policy

Reviewer #4: No

---

## [Editor Report · Acceptance letter]

PONE-D-25-20067R2

PLOS One

Dear Dr. Kaiser-Graf,

I'm pleased to inform you that your manuscript has been deemed suitable for publication in PLOS One. Congratulations! Your manuscript is now being handed over to our production team.

Kind regards,

on behalf of

Director Junzheng Yang

Academic Editor

PLOS One